# State of Health Estimation Method for Lithium-Ion Batteries via Generalized Additivity Model and Transfer Component Analysis

**Mingqiang Lin** [1,2,*]**, Chenhao Yan** [1,2] **and Xianping Zeng** [3,*]

1    School of Advanced Manufacturing, Fuzhou University, Jinjiang 362200, China
2    Quanzhou Institute of Equipment Manufacturing, Fujian Institute of Research on the Structure of Matter, Chinese Academy of Sciences, Jinjiang 362200, China
3    School of Aerospace Engineering, Xiamen University, Xiamen 361005, China
*    Correspondence: kdlmq@fjirsm.ac.cn (M.L.); zengxp@stu.xmu.edu.cn (X.Z.); Tel.: +86-150-5951-0022 (X.Z.)

**Abstract:** Battery state of health (SOH) is a momentous indicator for aging severity recognition of lithium-ion batteries and is also an indispensable parameter of the battery management system. In this paper, an innovative SOH estimation algorithm based on feature transfer is proposed for lithium-ion batteries. Firstly, sequence features with battery aging information are sufficiently extracted based on the capacity increment curve. Secondly, transfer component analysis is employed to obtain the mapping that minimizes the data distribution difference between the training set and the test set in the shared feature space. Finally, the generalized additive model is investigated to estimate the battery health status. The experimental results demonstrate that the proposed algorithm is capable of forecasting the SOH for lithium-ion batteries, and the results are more outstanding than those of several comparison algorithms. The predictive error evaluation indicators for each battery are both less than 2.5%. In addition, satisfactory SOH estimation results can also be obtained by only relying on a small amount of data as the training set. The comparative experiments using traditional features and different machine learning methods also testify to the superiority of the proposed algorithm.

**Keywords:** lithium-ion batteries; state-of-health; transfer component analysis





## 1. Introduction

The lithium-ion battery has been attracting increasing preference since its extraordinary properties such as high energy density, long life, and no memory effect, and has been extensively used as the mainstream power source of electric vehicles [1]. Nevertheless, the accompanying irreversible health degradation characteristic poses a major challenge to battery management [2]. Recently, battery state of health (SOH) estimation has captured widespread attention due to it being tightly bound to the remaining useful life and accurate state of charge (SOC) estimation of batteries. Real-time and exact estimation of battery SOH is of great importance for automotive applications. Rendering accurate information about battery performance during the driving process of new-energy vehicles is helpful to manage energy distribution and prevent catastrophic accidents and can perform battery fault diagnosis for maintenance and replacement planning [3].

Currently, SOH estimation methods can be roughly classified into three categories, i.e., the direct measurement, model-based, and data-driven-based SOH estimation algorithms. The direct method measures the capacity by integrating the amount of electricity released during the whole discharge cycle. However, most batteries are in the state of partial charge and discharge in the actual working condition, the capacity estimation by calculating the charge between two SOCs over-depends on the accuracy of SOC estimation [4]. Model-based approaches include empirical model, equivalent circuit model (ECM), and physics-based model (PM). The empirical model method is adopted to fit the linear

model, exponential model, and polynomial model by continuously updating the measured or estimated parameters during the cycle [5], but it is commonly only suitable for specific aging modes and battery types. Because internal changes of the batteries are unknowable, empirical models have difficulty capturing the complex aging process. The ECM and PM methods reflect the characteristics of batteries by establishing battery models and estimate SOH by identifying specific model parameters [6]. However, due to complex operating conditions and coupling degradation mechanisms, it is difficult to achieve accurate and robust capacity estimates over the entire life of a battery. Compared with model-based methods, data-driven methods have been widely studied due to the superiority of model-free, high precision, and strong robustness [7].

The data-driven methods concentrated on mapping the relationships between the SOH and the feature vectors in the battery aging process. Since the capacity attenuation of the battery is closely related to its remaining service life, SOH is differentiated from the SOC estimation that relies mainly on the open circuit voltage (OCV) [8]. Researchers have no consensus on the recognized parameters that can reflect the SOH of a battery. Thus, the data-driven-based SOH estimation problem boils down to mining a signal-sensitive feature that maps SOH varieties and employing it to construct a predictor to estimate the battery SOH. The slope of the voltage curve and the voltage variation in a certain fixed time are frequently used feature extraction methods [9]. Moreover, the methods of extracting geometric characteristics from images by transforming the original measurement data have also fascinated growing predilection. Furthermore, incremental capacity (IC) peak value [10], differential thermal voltammetry (DTV) [11], and differential thermal capacity (DTC) [12] features have been demonstrated to be capable of capturing SOH aging characteristics preferably.

Classical machine learning regression algorithms such as support vector regression (SVR) [12,13], Gaussian process regression (GPR) [14], and neural networks and their variants [15–17] are also commonly used to mine the mapping relationship between features and SOH. Wang et al. [18] adopted broad learning system (BLS) to effectively reconstruct the model through incremental learning, shortening the training process and avoiding catastrophic forgetting. However, the traditional machine learning model used to establish the SOH estimator requires a certain number of samples with actual values, and the sample features and corresponding actual values are taken as the training set. While complicated experimental circumstances are required to obtain the real value during real-world applications, and it takes months or even years to collect a vast majority of cyclic data, the data obtained are fairly limited and precious. Mainstream machine learning algorithms cannot learn enough features when using a small sample training, which severely restricts the application and availability of the algorithm.

Model-based transfer learning (TL) can transfer tasks from the source domain to the target domain and is commonly used to dispose of small-sample learning problems. Deng et al. [19] grouped multiple batteries according to the capacity attenuation rate, and each group was provided with a reference battery. A long short-term memory (LSTM) neural network model was established for the reference battery, and the reference model was fine-tuned by substituting part of the target battery data to obtain the SOH estimation model of the target battery. Wang et al. [20] presented an improved transfer learning SOC estimation algorithm based on a gated recurrent unit (GRU) model. By pre-training the GRU model in the source domain, the GRU hidden unit structure can be enhanced, and the weight parameters of the source domain were transferred to the GRU model of the target battery. Experimental results showed that the proposed improved GRU-based transfer learning method performed well on small samples. Shu et al. [21] combined the LSTM network and fine-tuning strategies-based TL model to establish the SOH cell mean model (CMM) using part of the training data. To evaluate the SOH inconsistency between batteries, the LSTM model was adopted as the cell difference model (CDM), and the minimum estimate of CDM was identified to determine the SOH of the battery pack, thus realizing the model migration.

Because of a wide variety of batteries and complex working conditions (changes in temperature, current, and other working environments), different internal electrochemical compositions, and complex external environmental factors, even the aging curves of the same type batteries are dissimilar. Thus, the distribution difference of different battery aging data varies greatly, the existing training sample sets are not necessarily suitable for the newly generated data, and the estimator trained on this basis struggles to meet the requirement. The model built with one battery dataset is difficult to generalize to other battery datasets.

The goal of feature-based transfer learning is to search for an optimally shared feature subspace, narrowing the distribution difference between the source domain and target domain data on this feature subspace, while preserving the feature mapping of its internal attributes to the greatest extent. Using the domain training model after feature transformation is essentially equivalent to increasing the training data to enhance the generalization ability of the model. Blitzer et al. [22] presented a method of structural correspondence, identifying pivot features that frequently appear in the source domain and a target domain and using these pivot features to establish cross-domain feature correspondence. Wang et al. [23] adopted structured sparsity-inducing norms to discover correlations between tasks during multi-task learning and then improved the prediction performance by sharing characteristics among related tasks. The effective measurement of distribution differences between domains is a critical component in feature transfer learning. The main commonly used methods of measuring discrepancy between domains are the Bregman distance [24], the entropy-based Kullback–Leibler (KL) divergence [25], and the maximum mean discrepancy (MMD) [26]. Because Bregman distance adopts the gradient descent method to solve the objective function, it requires a large amount of computation. KL divergence is frequently used in measuring the similarity between probability distribution functions, which requires continuous prior probability density estimation. Compared with Bregman distance and KL divergence, the measurement calculation of MMD is comparatively simple, straightforward, and understandable, and has been attracting increasing preference in feature transfer learning.

Motivated by the aforementioned limitations, this study focuses on developing an innovative SOH estimation algorithm based on feature transfer learning. Sequence features that are highly sensitive to SOH changes are sufficiently extracted from IC curves, and the distribution difference between the features of the source domain and target domain in a shared feature subspace is minimized by transfer component analysis (TCA) [27], while the internal attributes of the original data are maintained. The newly obtained characteristics of the source and target domains are input into a generalized additive model (GAM) to estimate the SOH of the battery, and then experiments on multiple batteries are conducted to substantiate the proposed transfer learning technique.

The layout of this paper is as follows: Section 2 introduces feature extraction methods of lithium-ion batteries. Section 3 describes the SOH estimation algorithm in detail. Experimental results and analysis are made in Section 4, followed by conclusions summarized in Section 5.

## 2. Feature Extraction

### 2.1. Definition of SOH

The cyclic aging data of lithium-ion batteries in this paper are obtained from the Oxford battery degradation dataset [28]. This dataset contains aging data of eight Kokam pouch batteries with a nominal capacity of 740 mAh, noted as cell 1 to cell 8. The dimensions of batteries are $58.5 \times 33.5 \times 5$ mm, and the model number is SLPB533459H4. The negative electrode material of batteries is graphite, and the positive electrode material is LiMO2 (where M represents a combination of Ni, Mn, and Co, commercially known as NMC). The cells were all tested in a thermal chamber at 40 degC. The current, voltage, and surface temperature of batteries were measured and recorded by a Bio-logic MPG-205 battery tester in every 100 aging cycles. The charging-discharging process of lithium-ion batteries

includes a 1C (740 mA) constant current-constant voltage (CC-CV) charge-discharge cycle and a 40 mA low current charge-discharge process.

The health status of lithium-ion batteries indicates the percentage of the maximum available capacity of the battery in the rated capacity, measuring the aging degree of the battery. With the aging of the battery, its capacity gradually diminishes. The battery SOH could be defined as

$$SOH = C_{current}/C_{initial} \qquad (1)$$

where $C_{current}$ represents the current capacity of the battery, $C_{initial}$ is the rated capacity, which is the initial capacity of the fresh battery. For a fresh battery, the initial SOH is assumed to be 100%. When the capacity attenuates to 80% of the initial capacity, the battery is considered to be invalid [29]. The capacity changes of eight cells are shown in Figure 1. Because of the inevitable changes of ambient temperature and mechanical stress, or internal battery failure in battery charging and discharging experiments, there are a certain number of invalid or missing values in the raw battery data. Prior to the feature extraction process, the original current, voltage, temperature, and other data in the battery dataset need to be cleaned. For voltage and current data, the moving average (MA) method was used to reduce the influence of sensor noise in sampling, and a robust locally weighted regression algorithm was used to smooth the temperature curve with poor robustness.

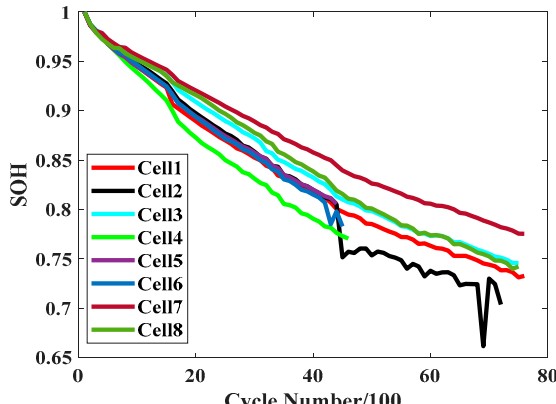

**Figure 1.** The capacity degradation curves of eight cells in the Oxford dataset.

*2.2. ICS Feature Vector Extraction*

IC analysis was originally derived from research by Thompson in the field of materials science in 1979 [30]. Later in the 1990s, Dahn adopted it to investigate carbon materials in lithium batteries [31]. In the study of the capacity degradation mechanism of lithium-ion batteries, IC represents the battery capacity increased at a continuous voltage increment. In the constant current charging mode, the calculation formula of IC is

$$IC = \frac{dQ}{dU} = I \cdot \frac{dt}{dU} \qquad (2)$$

where $Q$ denotes the capacity, $U$ is the voltage, and $t$ represents the sampling time.

SOH estimation needs to extract the characteristics reflecting battery degradation from the IC curve, usually adopting geometric or statistical features, such as the peak value of the IC curve or sample entropy [32]. However, as shown in Figure 2, in addition to the peak value, there are still fluctuations related to the aging degree in the IC curve, thus it is extremely vulnerable to noise interference if the IC peak value is adopted as a feature alone. To address this issue, the values of a segment of the IC curve are sampled as characteristics, and then the mapping relationship between the IC curve and SOH is established. For the given sampling step $\Delta U$ and voltage interval $[U_1, U_h]$, the sampling point $[U_1, U_1 + \Delta U, U_1 + 2\Delta U, \ldots U_1 + n\Delta U]$ can be determined, and then the ICS feature vector $h = [IC_1, IC_2, \ldots IC_n]$, with $n = [(U_h - U_1)/\Delta U]$ can be extracted, where ICS

means the IC feature vector with the length of $n$ containing the IC peak value. After analyzing the charging curves of each battery in the Oxford dataset, samples were taken at intervals of 0.01 V between 3.75 V and 4.04 V, and the ICS feature sequence with a length of 20 was extracted.

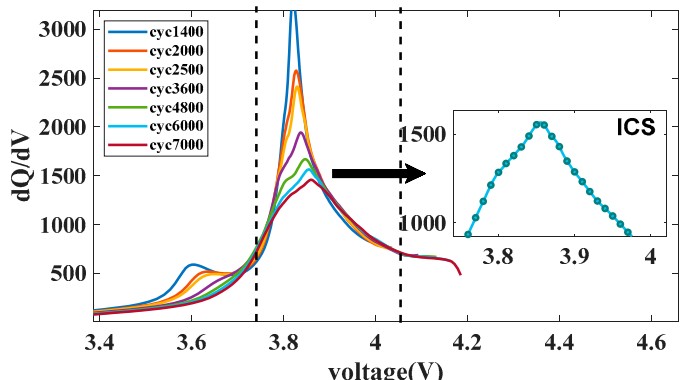

**Figure 2.** The incremental capacity curves of cell 1 in the Oxford dataset.

### 2.3. ICS Feature Transfer

A battery in the Oxford dataset is randomly selected as the source domain $D_s$, and $D_s = \{X_S, Y_S\}$, where $X_S$ is the ICS feature sequence of the battery in the source domain, and $Y_S$ is the corresponding actual SOH value. Suppose that the battery to be predicted is the target domain $D_T$, and $D_T = \{X_T\}$, where $X_T$ is the ICS feature sequence of the target domain, and both $X_S$ and $X_T$ are $D$-dimension feature data. Insufficient training samples will result in poor SOH estimation accuracy, which may be attributed to the data distribution difference between source and target domains, i.e., $P(X_S) \neq P(X_T)$. In this paper, the TCA method is employed to combine MMD with principal component analysis (PCA) to seek out a shared feature representation between adjacent domains. That is, assuming that there is a nonlinear feature mapping $\phi$, the TCA method can make the marginal probability distribution between the two domains in the mapped new feature subspace as consistent as possible, i.e., satisfy $P(\phi(X_S)) \approx P(\phi(X_T))$.

To calculate the aforementioned feature mapping $\phi$, the discrepancy between the source and target domains is measured by means of the MMD approach shown below

$$dist(D_s, D_T) = \left\| \frac{1}{n_S} \sum_{x_{Si} \in X_S} \phi(x_{Si}) - \frac{1}{n_T} \sum_{x_{Tj} \in X_T} \phi(x_{Tj}) \right\|_H^2 \qquad (3)$$

where $n_S$ is the number of aging battery samples in the source domain, and $n_T$ is the number of aging battery samples in the target domain. $\|\cdot\|_H$ represents the norm of reproducing kernel Hilbert space.

However, using raw data directly to minimize the objective function will bring about complicated $\phi$ calculation. For simplifying the implementation procedure and reducing the calculational cost, the TCA method introduces the following nuclear matrix $K$ and coefficient matrix $L$:

$$K = \begin{bmatrix} K_{S,S} & K_{S,T} \\ K_{T,S} & K_{T,T} \end{bmatrix} \in R^{(n_S + n_T) \times (n_S + n_T)} \qquad (4)$$

$$L_{ij} = \begin{cases} \frac{1}{n_S n_S} (x_i, x_j \in D_S) \\ \frac{1}{n_T n_T} (x_i, x_j \in D_T) \\ \frac{1}{n_S n_T} (otherwise) \end{cases} \qquad (5)$$

where $K_{S,S}, K_{T,T}, K_{S,T}$ and $K_{T,S}$ are the kernel matrices of the source domain, target domain, and cross-domains respectively, and $K(i,j) = [\phi(x_i)^T \phi(x_j)]$. $dist(D_s, D_T) = tr(KL)$, where $tr(\cdot)$ denotes the trace of the matrix.

In addition, dimensionality reduction theory is also adopted in the TCA method. Firstly, the kernel matrix $K$ is decomposed into $K = (KK^{-1/2})(K^{-1/2}K)$. The low-dimensional matrix $W' \in R^{(n_S+n_T)\times m}$ ($m \ll n_S + n_T$) is defined, and the kernel mapping is transformed to the $m$-dimensional space.

Then, $K$ is replaced with the empirical kernel $K' = (KK^{-1/2}W')(W'K^{-1/2}K) = KWW^TK$, where $W = K^{-1/2}W'$. Thus, $dist(D_s, D_T) = tr(K'L) = tr(W^TKLKW)$ is obtained.

Moreover, the regularization term $tr(W^TW)$ is introduced in the process of minimizing the objective function to maintain the variance of the sample data to the maximum extent to ensure that the data characteristics are preserved after transformation. Assuming that the variance of the data is $W^TKHKW$, where $H = I - (1/(n_S + n_T))qq^T$ is the central matrix, and $q \in R^{n_S+n_T}$ is the column vector whose elements are 1.

Finally, the objective function of the TCA algorithm is as follows

$$\begin{cases} \min\limits_{W} tr(W^TKLKW) + \beta tr(W^TW) \\ s.t. \quad W^TKHK^TW = I \end{cases} \tag{6}$$

where $\beta$ is a tradeoff factor. Therefore, the optimal mapping matrix $W$ is obtained by solving the above equation to implement the mapping of the ICS feature space of the source and target domains.

## 3. Methodology of SOH Estimation

### 3.1. GAM Method

The GAM method [33] was proposed by Trevor Hastie and Tibshirani in 1990. GAM is based on the sum of the generalized linear model and additive model. The application of this model can not only flexibly analyze the relevant parameters but also directly deal with the relationship between the response and multiple independent variables. Furthermore, the independent variables that have a complex nonlinear correlation with response are fitted into the model in the form of a summation of different functions. Suppose $x_i$ is the $i$-th feature of the sample and the number of features is $p$, then the mathematical expression of GAM is

$$g(\mu) = \alpha + \sum f_i(x_i) + \varepsilon \tag{7}$$

where $\mu$ is the expectation of, i.e., $\mu = E(Y|x_1, \ldots, x_p)$. $g(\cdot)$ is the connection function, and the identity function is employed to describe the regression problem in this paper, that is, $g(x) = x$. $\alpha$ is the intercept, and $\varepsilon$ is the error term, which is independent of the independent variables and obeys $N(0, \sigma^2)$. $f_i(x_i)$ is a smooth function of the predictor variable $x_i$, which is more adaptable than in the linear model. The B-spline function with the outstanding fitting ability is selected as the smooth function $f_i(x_i)$ in the experiment.

In the GAM method, the relationship between the independent variable and the smoothing function is linear, while the relationship between the dependent variable and the smoothing function is nonlinear. Assuming that each decomposition function in the model is additive and smooth, the data analysis based on the model does not require the linear assumption of the relevant independent variables in advance, and the dependent variables can be adopted in various exponential distribution forms. In the process of GAM training, a non-parametric method and additive hypothesis are employed, that is, the model can effectively explore the nonlinear relationship in the prediction function, enhancing the adaptability and flexibility of the algorithm.

### 3.2. TCA-Based SOH Estimation Methodology

In this paper, cell 8 in the Oxford dataset was selected as the source domain, and cell 1 to cell 7 was successively selected as the target domain to substantiate the effectiveness of the feature transfer model. The new feature data after feature migration of source and target domains are represented by $X_S^{new}$ and $X_T^{new}$ respectively. Then $D_S^{new} = \{X_S^{new}, Y_S\}$ and $D_T^{new} = \{X_T^{new}, Y_T\}$ are input into the GAM regression model as the training set and

the test set, respectively, to predict the battery SOH, and the pre-transfer feature data are adopted to train the GAM for comparison. The algorithm flowchart is shown in Figure 3.

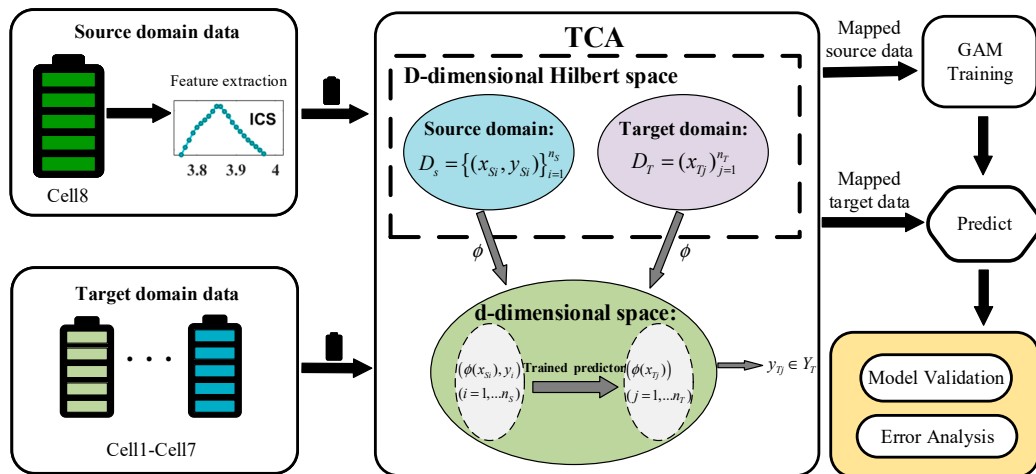

**Figure 3.** The flowchart of TCA-based SOH estimation algorithm.

## 4. Experiments and Analysis

To evaluate the performance of the estimation method, the mean absolute error (MAE) and the root mean squared error (RMSE) were opted as evaluation indicators. The definition of two metrics are shown in Equations (8) and (9).

$$MAE = \frac{1}{N}\sum_{i=1}^{N}|y_i - \hat{y}_i| \tag{8}$$

$$RMSE = \sqrt{\frac{1}{N}\sum_{i=1}^{N}(y_i - \hat{y}_i)^2} \tag{9}$$

where $\hat{y}_i$ and $y_i$ denote the predicted and actual values, respectively, $N$ is the number of test samples.

The predicted results of battery SOH before and after feature transfer are shown in Figure 4. The MAE and RMSE results are also calculated in Table 1.

As can be seen in Figure 4 and Table 1, without using the TCA method, the model trained using samples from one battery contained in the Oxford dataset made poor predictions for the other seven batteries. In the middle and later stages of the aging cycle, the predicted results of each battery are significantly different from the real values, and the overall forecast curve fluctuates greatly, with RMSE values all greater than 2%. However, the SOH prediction effect of each battery was significantly improved by using the TCA algorithm; the results are very close to the reference value for all operating cycles. From Table 1, it can be observed that the MAE values are all within 2%, especially the MAE values of cells 1, 3, and 7 are all within 1%. It is noted that the prediction results of cell 2 are worse than those of other batteries, probably since cell 2 experienced two capacity drops during the charge-discharge cycle. However, after using feature transfer learning, the MAE error of the cell 2 is reduced from 7.37% to 1.72%, and the RMSE value is reduced from 10.86% to 2.31%. The RMSE value of other cells are all within 2%. The results manifest the feasibility of the proposed algorithm.

To further verify the effectiveness of the proposed algorithm, the data of cell 8 were still adopted as the training set, and traditional IC features were extracted to train four commonly used machine learning models, namely, SVR, neural network (NN) [29], linear regression (LR) [34], and convolutional neural network (CNN) [35], to compare with the algorithm proposed in this paper. The SOH prediction effect of each model on the aging

curve of cell 1 to cell 7 is shown in Figure 5. The MAE and RMSE results of SOH estimation results of different models are also given in Table 2.

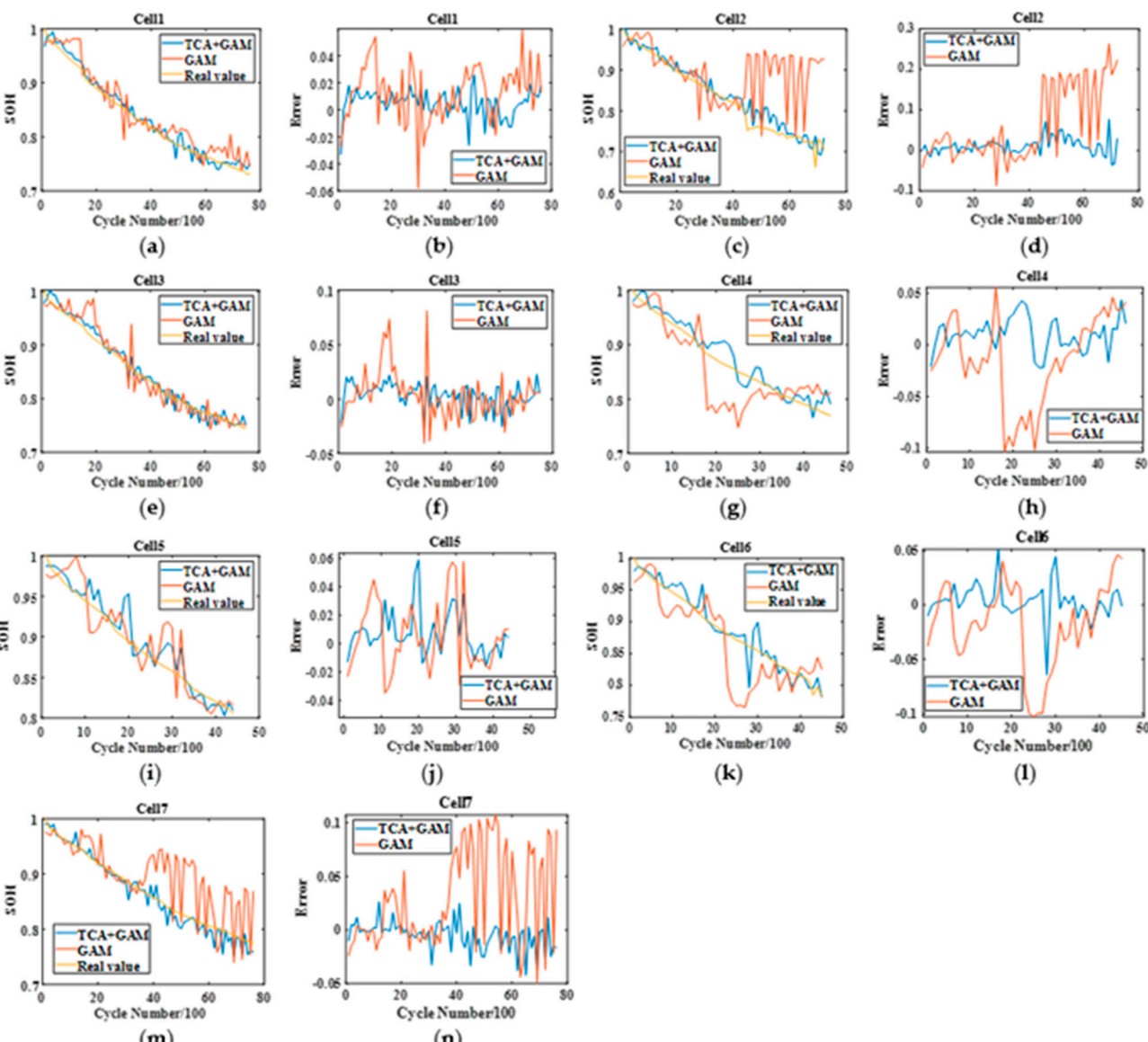

**Figure 4.** SOH estimation results before and after feature transfer on the Oxford battery dataset: (**a,c,e,g,i,k,m**) are the SOH prediction curves of cell 1 to cell 7; (**b,d,f,h,j,l,n**) are the predicted error curves of cell 1 to cell 7.

**Table 1.** MAE and RMSE of SOH estimation results before and after feature transfer.

| Cell | MAE | | RMSE | |
|---|---|---|---|---|
| | **TCA + GAM** | **GAM** | **TCA + GAM** | **GAM** |
| Cell 1 | 0.92% | 1.95% | 1.12% | 2.41% |
| Cell 2 | 1.72% | 7.37% | 2.31% | 10.86% |
| Cell 3 | 0.93% | 1.54% | 1.17% | 2.22% |
| Cell 4 | 1.55% | 3.47% | 1.90% | 4.46% |
| Cell 5 | 1.16% | 1.95% | 1.74% | 2.48% |
| Cell 6 | 1.10% | 3.29% | 1.73% | 4.30% |
| Cell 7 | 0.97% | 3.97% | 1.42% | 5.28% |

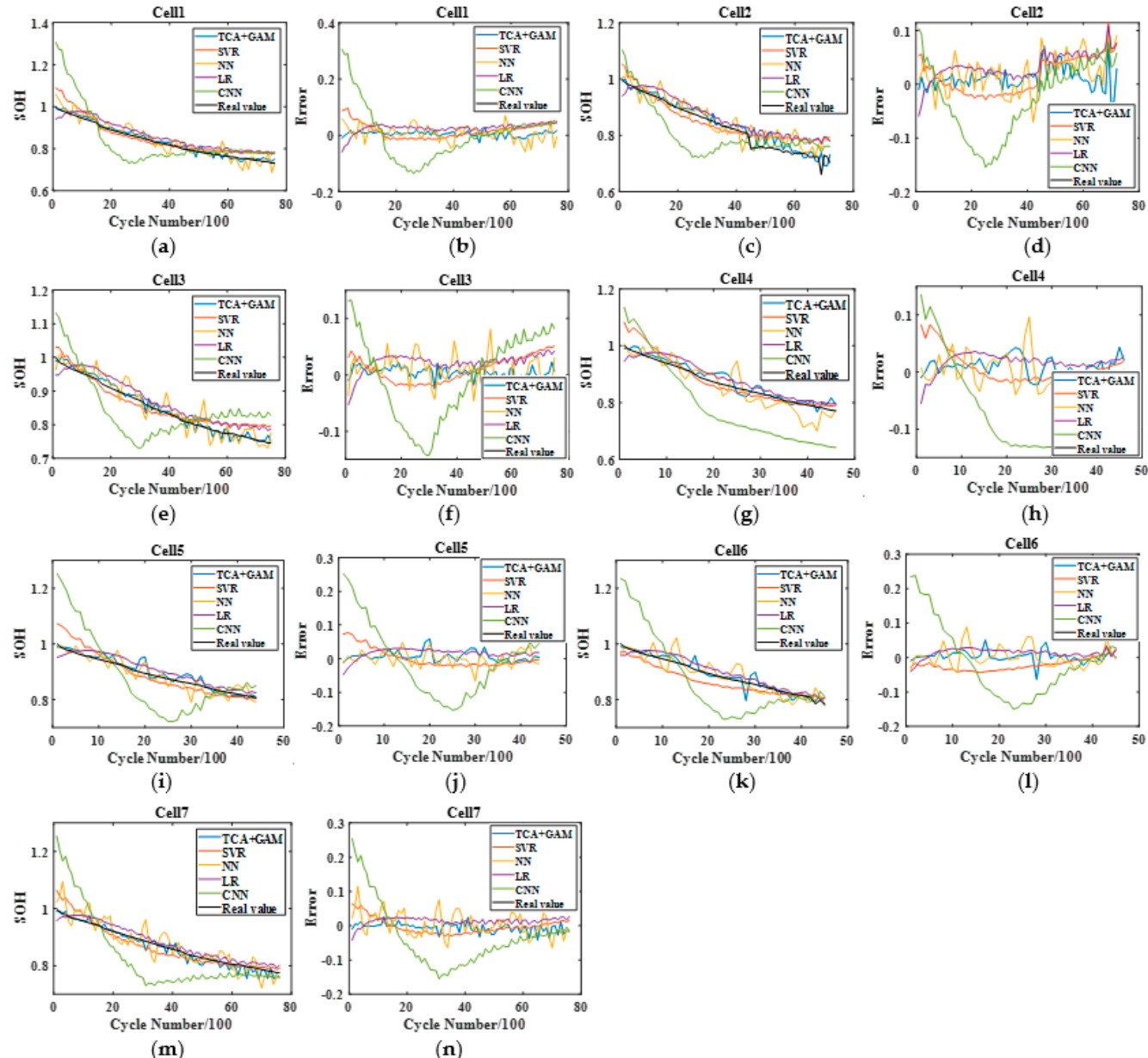

**Figure 5.** SOH estimation results of different models on the Oxford battery dataset: (**a,c,e,g,i,k,m**) are the SOH prediction curves of cell 1 to cell 7; (**b,d,f,h,j,l,n**) are the predicted error curves of cell 1 to cell 7.

**Table 2.** MAE and RMSE of SOH estimation results of different models.

| Cell | MAE | | | | | RMSE | | | | |
|---|---|---|---|---|---|---|---|---|---|---|
| | TCA + GAM | SVR | NN | LR | CNN | TCA + GAM | SVR | NN | LR | CNN |
| Cell 1 | 0.89% | 2.34% | 2.24% | 2.98% | 7.03% | 1.06% | 3.12% | 2.83% | 3.14% | 9.65% |
| Cell 2 | 1.74% | 3.09% | 3.26% | 3.71% | 5.97% | 2.31% | 3.73% | 4.01% | 4.24% | 7.38% |
| Cell 3 | 0.92% | 2.03% | 2.06% | 2.57% | 6.49% | 1.15% | 2.41% | 2.61% | 2.73% | 7.52% |
| Cell 4 | 1.53% | 1.95% | 2.90% | 2.15% | 10.52% | 1.88% | 2.88% | 3.67% | 2.34% | 11.25% |
| Cell 5 | 1.16% | 2.19% | 1.85% | 2.00% | 8.51% | 1.73% | 2.85% | 2.34% | 2.18% | 10.65% |
| Cell 6 | 1.08% | 2.48% | 2.25% | 1.68% | 8.38% | 1.71% | 2.77% | 2.98% | 1.91% | 10.48% |
| Cell 7 | 0.97% | 1.74% | 2.81% | 1.58% | 7.98% | 1.42% | 2.23% | 3.44% | 1.73% | 9.62% |

The experimental results show that the SVR, NN, LR, and CNN models trained on one cell have unsatisfactory predictive effects on the other seven cells due to the distribution difference between batteries. CNN algorithm has a poor fitting effect when training based on small samples, while the inconsistency of data distribution between the training set and the test set further increases the prediction error, with MAE and RMSE values both above 5%. The NN method can capture the general tendency of battery aging but provide low robustness, and the overall prediction curve is more volatile. SVR and LR models rely on outstanding small sample fitting ability to improve the smoothness of the predicted curve compared with the NN method, but the predicted results in the early stages of the aging cycle are significantly deviated from the real values, still unable to accurately track the aging status of the battery within the whole life cycle. Therefore, the traditional regression models cannot capture the dynamic aging characteristics of the battery based on small training samples, with a poor generalization performance of the trained model.

Since the GAM is based on a simple additive model, with a short time-consuming model training, and better results can be obtained without feature screening and normalization. GAM has a lower model complexity in comparison to other methods. In the case of a small number of training set samples, even if cell 1 to cell 7 with more samples are used as the test set, the features processed by the TCA method can still satisfactorily map the relationship between the features and the SOH value, and the overall predicted curves have a higher consistency and smoothness with the real values. The above-mentioned results validate that a small amount of experimental data can be taken full advantage of through feature transfer learning. It is further illustrated that the proposed algorithm has strong engineering practical significance.

## 5. Conclusions

An innovative SOH estimation algorithm based on feature transfer learning was proposed for lithium-ion batteries. Firstly, sequence features with battery aging information were sufficiently extracted from IC curves. Secondly, the TCA method was employed to obtain the mapping that minimizes the data distribution difference between the training set and the test set in the shared feature space. Finally, the GAM was investigated to estimate the SOH of the battery. Besides, comparative experiments on the Oxford battery degradation dataset were conducted to substantiate the effectiveness of the proposed model. Cell 8 in the Oxford dataset was adopted as the training set, and the other seven batteries were successively selected as test sets. The experimental results showed that the SOH prediction effect of each battery was significantly improved by using the TCA algorithm; the results are very close to the reference value for all operating cycles, and the maximum MAE and RMSE values for all cells are less than 2.5%. In addition, traditional IC features were extracted to train four commonly used machine learning models, i.e., SVR, NN, LR, and CNN, to compare with the algorithm proposed in this paper. The results also showed that the proposed method can better capture the dynamic characteristics of battery aging and provide higher accuracy and robustness of SOH estimation results in comparison to other methods. In addition, satisfactory SOH estimation results can also be obtained by only relying on a small amount of data as the training set, which makes it ideal for engineering applications.

**Author Contributions:** Conceptualization, M.L. and C.Y.; methodology, M.L. and C.Y.; software, C.Y.; validation, C.Y. and X.Z.; formal analysis, M.L. and X.Z.; investigation, C.Y.; resources, M.L.; data curation, C.Y.; writing—original draft preparation, C.Y. and X.Z.; writing—review and editing, M.L. and X.Z.; visualization, X.Z.; supervision, M.L.; project administration, M.L.; funding acquisition, M.L. All authors have read and agreed to the published version of the manuscript.

**Funding:** This research was funded by the National Natural Science Foundation of China (Grant number 62203423), and Fujian Provincial Natural Science Foundation (Grant number 2022J01504).

**Data Availability Statement:** Not applicable.

**Conflicts of Interest:** The authors declare no conflict of interest.

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
