# Peer review of "State of Health Estimation Method for Lithium-Ion Batteries via Generalized Additivity Model and Transfer Component Analysis"

_wevj, doi:10.3390/wevj14010014_

Round 1

Reviewer 1 Report

The submitted manuscript introduced an innovative SOH estimation algorithm based on feature transfer learning. Sequence features with battery aging information are sufficiently extracted from IC curves, and the distribution difference between the features of the source domain and target domain in a shared feature subspace is minimized by TCA method. Then the GAM is investigated to estimate the SOH of the battery. In addition, the comparative experiments on the Oxford battery degradation dataset are conducted to substantiate the effectiveness of the proposed model. Nevertheless, some details of the method introduced in this paper need further explanation. Several issues should be resolved before publication.

1. How to determine the parameter n in the feature extraction described in subsection 2.2, and how to influence the experimental results.

2. The authors should point out the specific voltage sampling interval in the experiment part.

3. The type of lithium battery used in the Oxford dataset, ternary lithium or lithium iron phosphate should be pointed out. Besides, the working conditions of the Oxford data are used should be also indicated.

4. How to obtain the capacity in the SOH calculation formula.

5. The error plots in Figure 4 have no legend.

6. What kind of smoothing function is adopted in the GAM method should be clarified.

7. The text font or size in all figures should be consistent and the overall appearance should be achieved as far as possible. For example, the font for the sub-box headings in Figure 2 can be bold and uniform. Some of the vertical fonts in Figures 3 and 4, such as “SOH” are distorted.

8. The authors should describe the comparative methods in more detail or supplement the corresponding references.

Reviewer 2 Report

The manuscript develops a feature transfer-based algorithm to estimate the SOH of lithium-ion batteries. But some improvements need to be made before it can be accepted for publication in WEVJ. The comments for improving the manuscript are listed as follows:

1. The quantitative accuracy results should be given in the abstract.

2. The overview of existing SOH estimation methods in the introduction is not comprehensive enough, and further supplements are needed.

3. The sampling interval delt U in subsection 2.2 should be illustrated in the text.

4. The authors should carefully check the content of the article to avoid the trouble caused by carelessness. For example:

(1) Some of the fonts in Figures 3 and 4 are not clear and deformed.

(2) Some plots in Figure 4 have no legend.

(3) Please unify the full text, and it is best to give the full name for the first time. Please check the full text.

(4) The format of equations (8) and (9) is not consistent with other equations.

5.  It is recommended to add a more detailed introduction to the battery dataset used.

6. The processing of the raw data in the dataset is not mentioned in the article.

7.  SOH is a key indicator reflecting the aging degree of batteries, and there is no uniform definition of the SOH. Why did the authors choose this method to calculate the true value of SOH?
